# Utilization of a Low-Cost Sensor Array for Mobile Methane Monitoring

**DOI:** 10.3390/s24020519

**Published:** 2024-01-14

**Authors:** Jonathan Silberstein, Matthew Wellbrook, Michael Hannigan

**Affiliations:** 1Department of Mechanical Engineering, University of Colorado at Boulder, 1111 Engineering Drive, Boulder, CO 80309, USA; 2Urban Labs, University of Chicago, 33 North LaSalle Street Suite 1600, Chicago, IL 60602, USA

**Keywords:** low-cost sensors, oil and gas well emissions, mobile monitoring, model calibration, quantification, screening tools

## Abstract

The use of low-cost sensors (LCSs) for the mobile monitoring of oil and gas emissions is an understudied application of low-cost air quality monitoring devices. To assess the efficacy of low-cost sensors as a screening tool for the mobile monitoring of fugitive methane emissions stemming from well sites in eastern Colorado, we colocated an array of low-cost sensors (XPOD) with a reference grade methane monitor (Aeris Ultra) on a mobile monitoring vehicle from 15 August through 27 September 2023. Fitting our low-cost sensor data with a bootstrap and aggregated random forest model, we found a high correlation between the reference and XPOD CH_4_ concentrations (r = 0.719) and a low experimental error (RMSD = 0.3673 ppm). Other calibration models, including multilinear regression and artificial neural networks (ANN), were either unable to distinguish individual methane spikes above baseline or had a significantly elevated error (RMSD_*ANN*_ = 0.4669 ppm) when compared to the random forest model. Using out-of-bag predictor permutations, we found that sensors that showed the highest correlation with methane displayed the greatest significance in our random forest model. As we reduced the percentage of colocation data employed in the random forest model, errors did not significantly increase until a specific threshold (50 percent of total calibration data). Using a peakfinding algorithm, we found that our model was able to predict 80 percent of methane spikes above 2.5 ppm throughout the duration of our field campaign, with a false response rate of 35 percent.

## 1. Introduction

Methane (CH_4_) is a colorless, odorless, flammable gas that comprises the majority of natural gas. Methane is typically used for power generation and heating, as well as fuel for vehicles powered via natural gas. Methane emissions represent the second largest contributor to climate change, following carbon dioxide (CO_2_) [1,2]. The annual mass of methane emissions is only three percent of that associated with CO_2_; however, the 100-year global warming potential of methane is 28 times greater than that of CO_2_, as the radiative forcing of methane is much greater than CO_2_ on a per mass basis [3]. Methane concentrations have increased dramatically from preindustrial levels of approximately 690 parts per billion (ppb) to current concentrations of 1850 parts per billion [4,5]. Elevated atmospheric methane concentrations contribute significantly to climate change and tropospheric ozone. Understanding the scope of methane sources may help forestall further increases in atmospheric methane.

Fugitive methane emissions arise from a variety of sources, including livestock, landfills, coal mining, and oil and natural gas wells. Within the United States, it is estimated between 50 and 65 percent of total methane emissions originate from anthropogenic activities [5]. Of the anthropogenic fraction of US methane emissions, oil and natural gas wells account for approximately one-third of the total flux [6]. Well sites are not distributed evenly throughout the US and are often situated in socioeconomically disadvantaged areas. This raises environmental justice concerns [7,8]. Leaking wells release not only methane but also other harmful volatile organic compounds (VOCs), including benzene, toluene, ethylbenzene, and xylene (collectively referred to as BTEX). Exposure to BTEX has been linked to increased incidences of asthma, cancer, and other serious cardiovascular impacts [9,10]. As the production of natural gas has expanded in recent decades with an increase in the number of wells and the proliferation of new drilling methods, the corresponding share of methane emissions attributable to oil and gas infrastructure has risen accordingly [11,12].

Accurately assessing methane concentrations is a necessary step in determining the emission rate of various point sources (such as oil and gas infrastructure) [13]. Commercially available tools used for fence line methane quantification typically rely on optical measurements to accurately measure methane concentrations. However, these tools are not without drawbacks, as they require costly equipment that can require extensive training and expertise to run correctly. Alternatively, over the past decade, studies have pioneered the use of low-cost sensors (LCS) to accurately quantify methane concentrations in lab studies [14,15,16], stationary deployments in urban areas [17,18,19], and fence line monitoring [20]. Other studies have leveraged the combination of low-cost sensors with machine learning methods to specify and predict individual VOC concentrations in stationary laboratory and field experiments with high fidelity [21,22]. Similarly, researchers have demonstrated the ability of machine learning algorithms for the classification and quantification of individual VOCs [22,23] and CH_4_ [24]. At a fraction of the cost of regulatory and research-grade monitors, LCS networks are used to supplement regulatory monitors by providing high-resolution spatiotemporal pollutant data and to inform local policies to best mitigate exposure.

Low-cost methane sensors typically fall within one of two classes: electrochemical (EC) and metal oxide (MOx) sensors. MOx sensors operate via an oxidation or reduction reaction when exposed to a target gas [25]. Target species interact with the sensor surface, resulting in the introduction or removal of free electrical charge in the semiconductive material [25]. This process changes the resistivity of the material, which is then measured and converted to a gas concentration. The resistivity of MOx sensors is highly dependent on both temperature and humidity, and sensor performance can degrade with time [26]. MOx sensors designed to react with CH_4_ as the target gas typically employ SnO_2_ deposited on an electrode and an Al_2_O_3_ substrate [27]. EC sensors operate via chemical reactions within a cell, which produce a current proportional to the concentration of the target gas [28]. However, these sensors are also susceptible to long-term degradation and effects from temperature and humidity changes [29,30]. The calibration of both MOx and EC LCS is often difficult due to a combination of sensor cross-sensitivity and performance degradation over time. Other commonly employed low-cost sensor technologies used for VOC assessment, such as photoionization detectors (PIDs), are not sensitive to changes in methane concentrations [31,32].

While the applications of low-cost sensor networks for oil and gas emission monitoring have been demonstrated in many studies [19,33,34,35], there is little research on the efficacy of these sensors for mobile applications. Tracking emissions over a large spatial boundary with transient sources requires an LCS platform capable of mobile measurements. In stationary studies, LCSs are deployed at either one or several fixed locations throughout the study duration, whereas for mobile studies, the sensor package is constantly moving. The use of LCS for mobile applications requires the consideration of several additional criteria. Sensors may be exposed to an emission plume for a period of multiple hours in stationary studies, whereas sensors in mobile monitoring applications may only be exposed to a source for several seconds. Sensors should be calibrated to collect data at a greater time resolution to account for the decreased time exposed to an emissions plume. Additionally, a subset of low-cost sensors interact with both VOCs and carbon monoxide, both of which are produced from vehicle exhaust. To avoid artifacts stemming from sensors responding to vehicle exhaust rather than methane emissions, sensors must be sited on the monitoring vehicle to minimize exposure to tailpipe emissions; while commercial monitors have been successfully employed to track these emissions, the cost of these instruments is often prohibitive [36]. A calibrated LCS device capable of accurately assessing CH_4_ concentrations may act as a low-cost alternative to costly research-grade equipment for the identification of CH_4_ spikes when coupled with an accurate peakfinding algorithm. To our knowledge, this study represents the first attempt to leverage LCS technology for mobile CH_4_ tracking from oil and gas infrastructure.

## 2. Materials and Methods

LCS data were collected over 16 days from 15 August 2023 to 27 September 2023. Data were collected using the XPOD monitoring platform—a low-cost sensor package employing commercially available sensors designed for air quality monitoring developed by the Hannigan Lab. Raw XPOD data were collected during monitoring every two seconds. Sensors employed by the XPOD monitoring platform to assess methane concentrations are shown in Table 1. Sensors were selected for this study based on a combination of price, sensing technology, and widespread usage in the relevant literature. Though selected sensors have differing manufacturer-prescribed sensing ranges and target gases, all sensors (with the exception of Alphasense VOC-B4, which, to our knowledge, has not been previously characterized in the literature) have been extensively studied and have been shown to correlate well with CH_4_ [16,33,37]. The XPOD monitor was placed within a University of Chicago emission monitoring vehicle (Appendix A). The average velocity of the vehicle near O&G facilities was several meters per second. All data collected from the XPOD and Aeris over the duration of this study were mobile data. The inlet of the XPOD was connected to the roof of the monitoring vehicle via 8 feet of inert Tygon tubing. Inlet air was pumped into the XPOD via a micro pump (Sensidyne) calibrated to a flow rate of 2.5 L per minute. Reference CH4 measurements were provided via a research-grade Aeris Ultra gas analyzer (Aeris Ultra, Project Canary, Denver, CO, USA). The inlet of the Aeris gas analyzer was placed adjacent to the XPOD inlet on the roof of the monitoring vehicle to minimize differences in the gas composition during deployment [38] (Appendix A). The response lags for pumped gas to enter both instruments were equal for the XPOD and Aeris Ultra.

Sampling occurred in the Julesburg Basin, a region comprising the area east of the Rocky Mountains in Colorado and Wyoming that extends to the western portion of Kansas and Nebraska. The Julesburg Basin produces both oil and natural gas from a combination of sand and shale formations [39]. Large-scale commercial oil and gas extraction within the Julesburg Basin have occurred since the early 1950s, resulting in a large number of legacy wells [39,40]. With the development of new extraction techniques, including the combination of horizontal drilling and hydraulic fracturing, new wells have continued to proliferate within the Julesburg Basin [41] (Appendix A). Methane leaks from O&G infrastructure are treated as a point source, and are often analyzed using a Gaussian plume dispersion model [42]. Many of these wells are in close proximity to large population centers within this region, including the cities of Greeley, Cheyenne, and Denver. A map of the Julesburg Basin and the spatial extent of our field monitoring are shown below in Figure 1.

XPOD CH_4_ concentrations were calculated using raw signal from sensors displayed in Table 1, as well as temperature and humidity. Reference and XPOD data were time-averaged in 15 s intervals using mean values over each period. To reconstruct CH_4_ concentrations from model variables, we applied multilinear regressions, random forest models (RF), and artificial neural networks (ANN), trained using Aeris reference measurements. We assessed the performance of these models using 2-fold cross-validation. Training and evaluation datasets for RF and ANN models were fit according to the methodology outlined in [43,44], with 80 percent of data used for training and 20 percent used for testing. Prior to fitting, XPOD sensor warmup periods were removed from deployment data, and the distribution of reference CH_4_ data was cleaned and then resampled using five concentration bins according to the methodology outlined in a study by Furuta et al. [37]. Following data binning, the training dataset comprised 18 h of data (corresponding to approximately 4300 data points), and the evaluation dataset comprised 4.5 h of data (corresponding to approximately 1100 data points). Resampling results in an improved balance among the number of experimental observations at different CH_4_ concentrations, at the expense of reducing the size of the overall dataset (Appendix A). The original and resampled reference CH_4_ distributions, as well as model fitting parameters on the original data distribution, are displayed in Appendix A.

### 2.1. Multilinear Regression Model

Multilinear regressions between reference Aeris CH_4_ concentrations and XPOD sensor signals, as well as temperature and humidity, are the simplest models to employ, as well as the easiest to interpret. Other authors have used these models during stationary deployments to accurately assess CH_4_ concentrations via LCS [17,41]. We produced multilinear regressions between reference CH_4_ from the Aeris Ultra monitor and the XPOD, calculated as
(1)$$\begin{array}{c} {\hat{y}}_{CH_{4}}\left(S_{2600},S_{2602},S_{2611C00},S_{2611E00},S_{AlphasenseVOC},S_{MQ4},T_{air},RH_{air}\right)=\\ \alpha_{0}+\alpha_{1}S_{2600}+\alpha_{2}S_{2602}+\alpha_{3}S_{2611C00}+\alpha_{4}S_{2611E00}+\alpha_{5}S_{AlphasenseVOC}+\alpha_{6}S_{MQ4}+\alpha_{7}T_{air}+\alpha_{8}RH_{air} \end{array}$$
where ${\hat{y}}_{CH_{4}}$ is the predicted CH_4_ concentration (in ppm), $S_{xx}$ represents raw sensor outputs for respective sensors, $T_{air}$ is the temperature of the air (in K), and $RH_{air}$ is the relative humidity of the air (in %).

### 2.2. Artificial Neural Networks

Recent studies of stationary low-cost sensor networks have shown that artificial neural network machine learning models are able to accurately translate sensor voltages to CH_4_ concentrations [19]. ANNs are composed of single units (neurons) ordered in a connected layer, with weights and an activation function applied to each neuron. Each layer of the ANN is connected to units in the previous layer. In our ANN, information is propagated forward through the network from the inputs, through hidden layers and bias functions to the output. For this model, the hyperbolic tangent function was chosen as the activation function. Neuron and bias weights in the ANN network were assigned randomly and iteratively adjusted to minimize a predetermined cost function. The number of hidden layer neurons were manually tuned to achieve optimal fitting performance. For our ANN, we employed a Bayesian regularization training function, which applies an additional term to the cost function that penalizes the network for increased complexity in order to help prevent overfitting. This regularization algorithm has been previously shown to perform well for regression applications independent of network architecture [19,45]. Other commonly used regularization functions, including Levenberg–Marquardt regularization and gradient decent regularization, displayed poorer fits than Bayesian regularization (Appendix A). More complex ANN architectures were not employed in this study, as more intricate ANN designs with additional hidden layers and neurons are likely to result in overfitting in smaller datasets [46]. A visualization of the ANN architecture is shown in Appendix A, and additional information regarding model hyperparameters and settings is included in Appendix A.

### 2.3. Random Forest Models

Random forests are a general classification of machine learning ensemble models consisting of several decision trees used to fit complicated data [47]. Random forests operate by creating an ensemble of decision trees fit on a training dataset, constructed from a random subset of predictor variables. Fitting parameters, including the number of leaves, the number of observed predictors included in the model, and sampling with replacement, were determined by minimizing the fitting error on testing data (Appendix A). Accordingly, the minimum leaf size in our random forest was set to five, and all eight parameters were sampled at each node. Data were sampled with replacement, and the prediction was generated by averaging the outputs of all trees. For this analysis, bootstrap aggregation (bagging) was employed for RF models due to the low dimensionality of our dataset.

### 2.4. Model Performance Evaluation

Optimal model parameters were selected by minimizing the root mean squared deviation (RMSD) between the reference and experimental measurements [48,49]. The RMSD consists of the sum of squared bias (SB), the difference in magnitude fluctuation (SDSD), and the lack of positive correlation multiplied by the standard deviation (LCS):(2)$$SB={\left(\bar{y_{CH_{4}}}-\bar{{\hat{y}}_{CH_{4}}}\right)}^{2}$$
(3)$$SDSD={\left(\sigma_{ref}-\sigma_{model}\right)}^{2}$$
(4)$$LCS=2\sigma_{ref}\sigma_{model}\left(1-\rho \right)$$
(5)$$RMSD=\sqrt{\left(SB+SDSD+LCS\right)}$$
where $\bar{y_{CH_{4}}}$ is the mean of the reference data, $\bar{{\hat{y}}_{CH_{4}}}$ is the mean of the model-predicted data, $\sigma_{ref}$ is the standard deviation of the reference data, $\sigma_{model}$ is the standard deviation of the model-predicted data, and ρ is the Pearson correlation coefficient between the model-predicted and reference data.

## 3. Results and Discussion

### 3.1. Calibration and Model Parameters

In developing a mobile CH_4_ model, we found a large variation in the efficacy of specific sensors in quantifying CH_4_. Alphasense’s electrochemical VOC sensor and MOx sensors designed with CH_4_ as a target gas displayed greater correlation with reference CH_4_ than general VOC MOx sensors (Figure 2). Relative humidity and temperature displayed significant correlations with reference CH_4_ concentrations over the duration of our study, which may be attributed to variable local meteorological conditions. All variables shown in Figure 2 were employed in calibration models during XPOD deployment to assess CH_4_ concentrations.

MLR and NN models were able to quantify longer-term changes in CH_4_ baseline but were unable to process rapid changes in CH_4_ from fugitive O&G leaks (Figure 3). RF models were able to quantify both short-term and longer-term fluctuations in CH_4_ signals (Figure 3). All three models displayed lower signal variability than the reference data at high CH_4_ concentrations due to the large contribution of baseline data during deployment. We ran each model on training data 100 times to minimize stochastic variation between runs of the same model, and selected the best-performing models within each model class for further analysis. Pre-binned data displayed no variation in RMSD from run to run as model parameters were fit to the same sample dataset. Additional information regarding model parameters and statistics can be found in Appendix A.

Prior to binning, model fits displayed high errors, as the RMSD for pre-binned data ($RMSD_{MLR}$ = 0.4189, $RMSD_{ANN}$ = 0.6917, and $RMSD_{RF}$ = 0.5023) was comparable to the variation between data points. The distribution of the reference methane concentrations (Appendix A) is heavily weighted toward baseline concentrations, as the majority of measurements occur at ambient conditions rather than an even distribution across the measured concentration spectrum. Applying data binning to reduce the inequality across the measured concentration gradient dramatically reduced fitting errors, as our applied calibration models provided a greater relative composition of higher concentration methane data.

Following data binning analysis of RMSD on testing data, it was shown that for NN models, a model consisting of a single hidden layer and 10 neurons minimized errors ($\mu_{RMSD}=0.4669$ ppm and $\sigma_{RMSD}=0.0737$ ppm). NN models displayed high variability as the number of neurons changed, indicating a sensitivity to tuning parameters. NN model variability may be attributed to the small sample size of the dataset and the select range of variables to alter. RMSDs for MLR models (($\mu_{RMSD}$ = 0.3652 ppm and $\sigma_{RMSD}$ = 0.0012 ppm) were lower than machine learning model configurations, but they displayed the lowest sensitivity to short-term variation in CH_4_, making these models poorly suited for mobile monitoring where CH_4_ spikes may last only several seconds. Larger NN models consisting of additional neurons better fit the training data, but they resulted in nonphysical interpretations of testing data due to overfitting, thus resulting in greater RMSDs. RF models most accurately assessed variation in short-term CH_4_ spikes and captured the overall trends in baseline variation. The RMSD on testing data showed a lower error for all RF models than any of the assessed NN models, and a similar magnitude of error to that of MLR models. RF models displayed significantly lower variability as the model inputs (tree number) were changed when compared to NNs. We attribute this diminished variability to the ensemble bagging process employed by RF models, which aggregates predictions from multiple trees to inform the final model prediction, thus reducing the weight of predictions from any single tree. Furthermore, the presence of outlier CH_4_ spikes and the high correlation between many of our sensors are well suited for random forest regression [47]. The optimal RF configuration consists of a forest comprising 100 trees ($\mu_{RMSD}$ = 0.3674 ppm and $\sigma_{RMSD}=0.0182$ ppm). This model was analyzed in further detail in the subsequent sections.

### 3.2. Evaluating the Impact of Additional RF Training Data

For regression applications, machine learning model performance often varies non-linearly with the amount of training data employed [50]; while reducing the total amount of training data in machine learning models can lead to overfitting, gains from including additional training data must be balanced by the cost of data collection [51]. We investigated the error between reference and model data for different percentages of total training data (Figure 4). We reduced data in all bins by percentages varying between 5 and 95 percent and ran 100 RF models on each reduced set of training data. Each 5 percent of binned data represents approximately 4 h of sampling before any preprocessing functions are applied. Between 5 and 50 percent of the total training data, the RMSD for testing data decreases, indicating that additional data points reduce error and further improve the RF model (Figure 4). A *t*-test analysis of adjacent data percentages (Appendix A) shows that the mean RMSDs are more likely to be statistically distinct for different amounts of data between 5 and 50 percent (Z1 in Figure 4) than between 50 and 100 percent (Z2 in Figure 4). With larger percentages of training data, additional data points no longer improve the RMSD, indicating that a precision XPOD sensor array may limit the predictive power of the RF model when more data points are used.

### 3.3. Assessing RF Model Sensor Performance

Due to the black-box nature of machine learning regression algorithms, it is often difficult to interpret which variables are contributing to model performance [52]. To qualitatively assess which variables are most relevant for RF model performance, we analyzed the distribution of predictor importance estimates for all model variables by running our chosen RF model 100 times, removing a specific sensor variable for each set of runs (Figure 5). We assessed the distribution of error values for our subset model and then subtracted the mean baseline error for the full RF model. Variables that, when removed, resulted in significant increases in model error have greater predictive importance than those that have a minimal impact. The sensor variables with the highest experimental correlations (Figure 2) with CH_4_ (Fig 2600, Fig2611-E00, Alphasense) have the greatest importance in our RF model, indicating sensor variables that are highly correlated with methane have greater predictive power. The removal of the Fig-2600 and Fig2611-E00 sensors dramatically increased model error by 25 and 30 percent, respectively. The MQ4 MOx sensor, which displayed a moderate correlation with the reference CH_4_, had a lower predictive importance than other sensors with similar correlation coefficients (2611-E00 and Alphasense). We hypothesize that the lower predictive importance of the MQ4 and Fig-2611 may be attributed to the high correlation between the MQ4 and Fig2611-E00 sensors (r = 0.93) and the Fig2611 and Fig2611-E00 sensors (r = 0.84). Permutations to the MQ4 signal may have diminished the influence on the error metric, as this sensor may provide redundant data to the model, with the weight of the data accordingly reduced. RF models, excluding Fig-2602, which displayed a negative correlation with CH_4_ over the course of the deployment, marginally reduced the RMSD when it was excluded from the RF model, indicating that this sensor may have contributed to overfitting during model calibration.

### 3.4. Utility of RF Model for CH_4_ Peakfinding

We further investigated the capability of our chosen RF model to assess short-term spikes in CH_4_ concentrations. CH_4_ spikes were determined via a one-dimensional peakfinding algorithm, whereby a peak was defined as a sample greater than its two nearest neighbors. We included additional prominence and magnitude constraints in the peakfinding algorithm to assess only the largest methane spikes. Reference CH_4_ spikes were defined as CH_4_ concentrations greater than 2.5 ppm with >0.5 ppm prominence. Throughout the monitoring campaign, there were a total of 20 peaks that met these criteria (Figure 6). The CH_4_ spike criteria for RF model data, which displayed lower variability at elevated CH_4_ concentrations when compared to the reference data, were adjusted accordingly. RF CH_4_ spikes were defined as points where model CH_4_ concentrations were greater than 2.05 ppm and model prominence was >0.1 ppm. Using the calibrated RF model, there were 27 peaks that fit these criteria (Figure 6). Of the 20 peaks defined by the reference CH_4_, 16 overlapped between reference and calibrated data, 3 peaks for reference CH_4_ displayed an increased model CH_4_ concentration below our target threshold, and 1 peak was missed by the RF model, indicating an accuracy of 80 percent for our RF model in determining CH_4_ spikes. Of the seven extra peaks predicted by the RF model, all but one occurred in regions with elevated CH_4_ baselines, as the RF model may have difficulty in predicting additional elevations to CH_4_ when baseline concentrations are raised. Increasingly sophisticated peakfinding algorithms, which employ local concentration and prominence thresholds rather than global values, may display better predictive value for assessing CH_4_ spikes.

## 4. Summary and Conclusions

Over a month-long deployment, we employed an array of LCS mounted on a mobile monitoring platform to reconstruct short- and longer-term fluctuations in CH_4_ concentrations stemming from fugitive oil and gas emissions. For mobile monitoring, specific MOx sensors targeting CH_4_ were able to quantify CH_4_ variations more accurately than general gas phase VOC sensors. Employing preprocessing functions to equitably sample CH_4_ concentrations across the full range of concentration space drastically improved fitting performance. Testing a wide range of models to fit our deployment data, we found that an RF model outperformed both ANN and MLR. RFs were able to capture longer-term variation in CH_4_ concentrations, as well as short-term spikes caused by fugitive emissions. Additionally, as the percentage of colocation data was reduced, the RF model performance did not significantly suffer until approximately half of the data were removed, indicating that, even in data-scarce environments, RF models may achieve high performance given a small parameter space from which to sample. Accordingly, even short-term field campaigns with LCS networks may be sufficient to achieve relatively high fidelity for CH_4_ measurements, assuming that measurements within the concentration space are well distributed.

Using our model, we were able to achieve similar error metrics when compared to other stationary LCS CH_4_ quantification studies [17]. Given the transient sampling environment in which our study occurred, our model may be less likely than stationary studies to be fit to specific local behavior and may be more generalizable, thus minimizing the risk of overfitting. However, in our study, we found the distribution of CH_4_ measurements to be heavily skewed toward baseline values, which may have caused our model to underpredict concentrations of CH_4_ spikes. Finally, the cost associated with generating data for our study was much greater than stationary monitoring, as the XPOD required active transportation to different field sites. In the future, extended field campaigns will need to be conducted to better understand and model longer-term seasonal fluctuations in CH_4_.

## Figures and Tables

**Figure 1 sensors-24-00519-f001:**
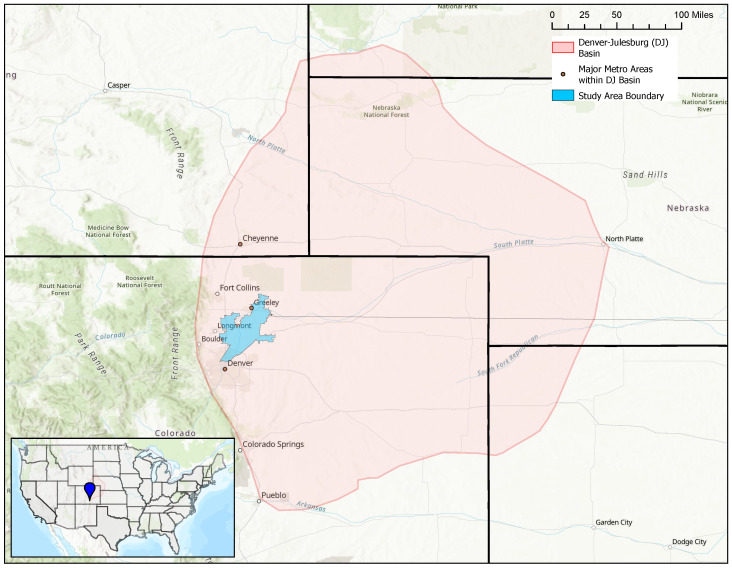
Map of study area. The pink-shaded area represents the Denver-Julesburg Basin, and the blue-shaded region represents the extent of our study.

**Figure 2 sensors-24-00519-f002:**
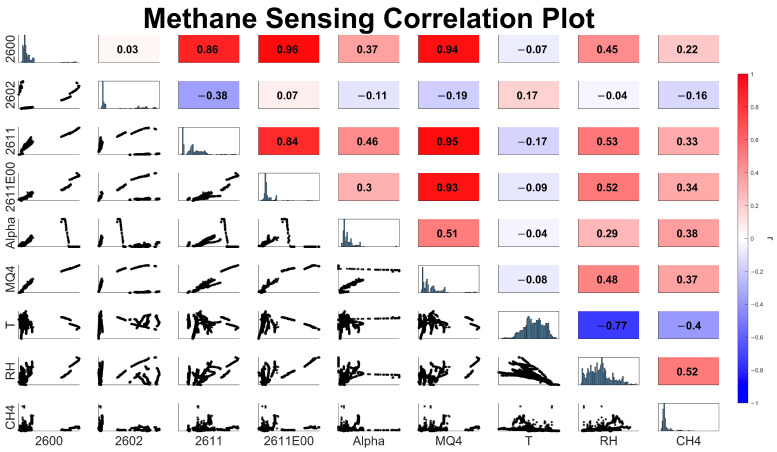
Scatter plot matrix of the raw sensor signal. The bottom triangle displays pair plots between individual variables, blue plots along the diagonal display distributions of each variable, and the upper triangle displays Pearson correlation coefficients for each variable pair.

**Figure 3 sensors-24-00519-f003:**
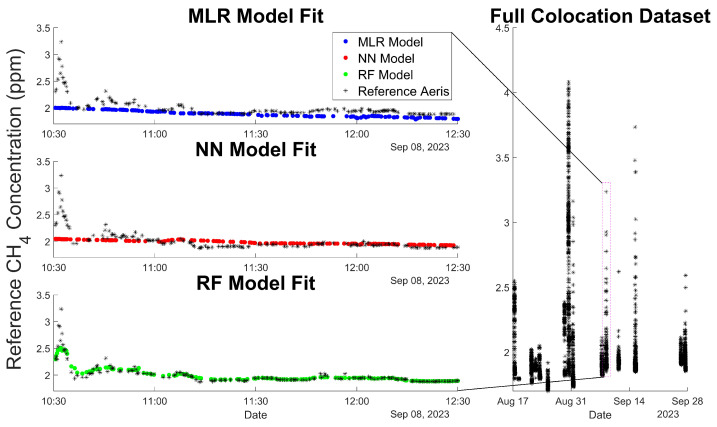
Model comparison over a single day of data acquisition. Individual models are displayed on the left-side plots, and the full Aeris reference dataset is displayed on the right-side time series. Additional model comparisons on August 30th are displayed in Appendix A.

**Figure 4 sensors-24-00519-f004:**
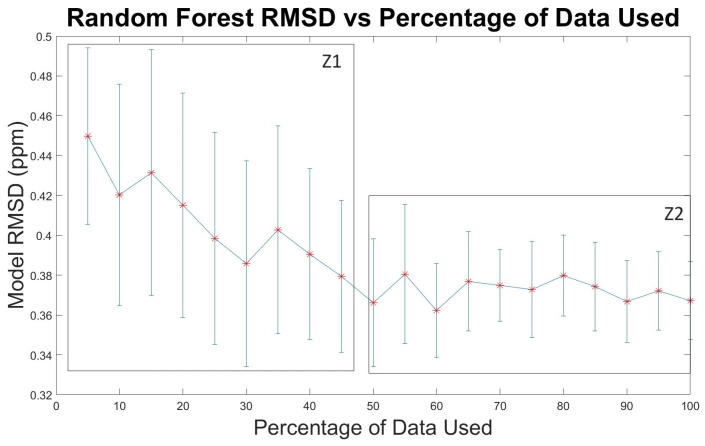
Error comparison for various percentages of training data run on 100 RF models. Error bars represent 1σ of RMSD.

**Figure 5 sensors-24-00519-f005:**
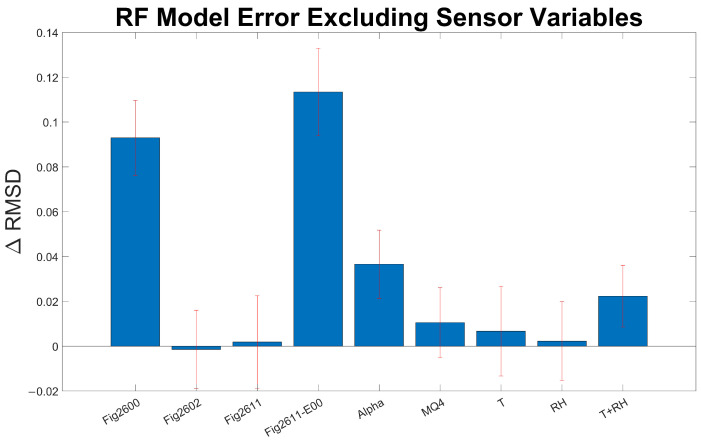
RF model ΔRMSD (ppm) excluding specific sensor signal over 100 runs. ΔRMSD (ppm) is calculated as the difference in RMSD between the sensor-excluded model and the base RF model. Error bars represent 1σ of ΔRMSD.

**Figure 6 sensors-24-00519-f006:**
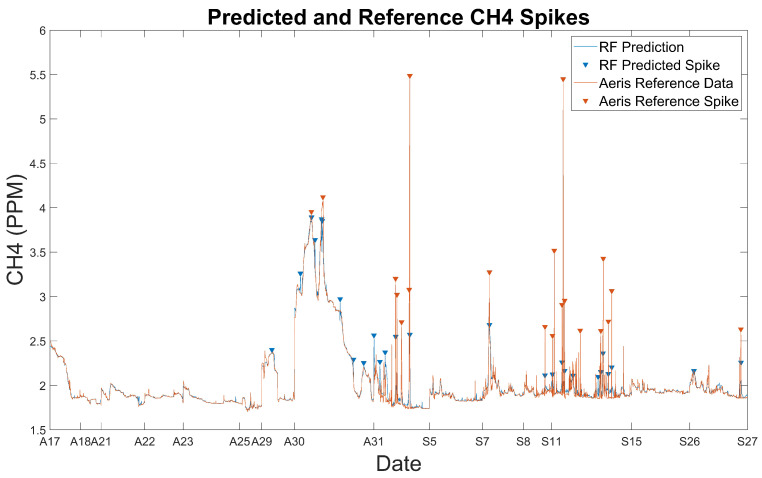
CH_4_ spike prediction using reference (orange) and calibrated model (blue) data.

**Table 1 sensors-24-00519-t001:** Sensors employed in mobile CH_4_ monitoring platform.

Sensor	Manufacturer	Target Gas	Sensing Range	Aprox. Cost (USD 2023)	Technology
Figaro 2600	Figaro Engineering (Osaka, Japan)	General VOC/air pollutants	Hydrogen 1–30 ppm	10	MOx
Figaro 2602	Figaro Engineering	General VOC/air pollutants	Ethanol 1–30 ppm	10	MOx
Figaro 2611-C00	Figaro Engineering	Methane	10,000–250,000 ppm	10	MOx
Figaro 2611-E00	Figaro Engineering	Methane	10,000–250,000 ppm	15	MOx
Alphasense VOC	Alphasense (Great Notley, Braintree, Essex, United Kingdom)	General VOCs	Gas dependent	150	EC
MQ4	Henan Hanwei Electronics (Zhengzhou, China)	Methane	200–10,000 ppm	5	MOx

## Data Availability

The datasets generated and analyzed in this study are provided in the Appendix A. Datasets include sensor values, reference Aeris values, and model fitting parameters.

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
