# Peer review of "Utilization of a Low-Cost Sensor Array for Mobile Methane Monitoring"

_sensors, 2024, doi:10.3390/s24020519_

Round 1

Reviewer 1 Report

Comments and Suggestions for Authors

The authors utilized a mobile sensor array to collect the gas emission data and different machine learning models were assessed to identify the peak concentration of methane. It is interesting research. However, there are still some issues which should be enhanced before acceptance.

 1.The literature review is not enough. The kind of gas detection method and gas sensors were not discussed in detail. The problems for current  gas detection methods were not  clarified clearly. For the researches about  gas detection with sensor array, there have been many reports, such as "SWCNTs-based MEMS gas sensor array and its pattern recognition based on deep belief networks of gases detection in oil-immersed transformers(https://doi.org/10.1016/j.snb.2020.127998) and Gas recognition method based on the deep learning model of sensor array response map(https://doi.org/10.1016/j.snb.2020.129349).The authors should present more discussions in this field.

2. What is the gas emission scenario for this research? The information about the gas emission source was not clear.

3. The details about the models used in the research were absent, e.g. the parameters selection for ANN and RF. The authors should add more information about it.

4. What is the difference between static sensor and mobile sensor? I think it is also an important for this issue. I did not find how to calibrate your sensors or collect standard samples for modeling. If the training samples were collected in the lab with static sensors, the difference between mobile and static conditions may have impact for concentration prediction. The authors should provide more discussion about it.

5. What about the sample size? The details about the samples both for training and test should be added in the manuscript.

Author Response

We appreciate the reviewer’s comments, which contribute to improving this manuscript. We have considered each comment carefully and given responses below.

(1)Comment: The literature review is not enough. The kind of gas detection method and gas sensors were not discussed in detail. The problems for current gas detection methods were not  clarified clearly. For the researches about  gas detection with sensor array, there have been many reports, such as "SWCNTs-based MEMS gas sensor array and its pattern recognition based on deep belief networks of gases detection in oil-immersed transformers(https://doi.org/10.1016/j.snb.2020.127998) and Gas recognition method based on the deep learning model of sensor array response map(https://doi.org/10.1016/j.snb.2020.129349)”.The authors should present more discussions in this field.

Response: We have added additional information and sources regarding the use of machine learning algorithms for the speciation and quantification of VOCs and methane to the introduction to further the literature review (Lines 53-58).

(2)Comment: What is the gas emission scenario for this research? The information about the gas emission source was not clear.

Response: We have added additional information about the emissions scenario and source (Lines 129-131).

(3)Comment: The details about the models used in the research were absent, e.g. the parameters selection for ANN and RF. The authors should add more information about it.

Response: We have added additional information about the parameters selection for ANN and RF models, as well as the methodology for choosing these parameters in section 2.2, 2.3, and supplementary tables.

(4)Comment: What is the difference between static sensor and mobile sensor? I think it is also an important for this issue. I did not find how to calibrate your sensors or collect standard samples for modeling. If the training samples were collected in the lab with static sensors, the difference between mobile and static conditions may have impact for concentration prediction. The authors should provide more discussion about it.

Response: We have defined the difference between static and mobile sensors in the introduction (Lines 80-83) and provided additional clarity regarding data collection (Lines 112-113).

(5)Comment: What about the sample size? The details about the samples both for training and test should be added in the manuscript.

Response: We have included the number of datapoints in testing and training (Lines 144-147).

Reviewer 2 Report

Comments and Suggestions for Authors

This work aims to use low cost sensors for mobile monitoring of CH4 with different statistical models for calibration and peakfinding. I have listed a couple of comments for authors to address for revision -

A very fundamental question - can authors justify why they don't use the same model's of sensors for the same gas detection? For example, for methane, they have Figora and Sparkfun which have a very large mismatch in the sensing dynamic ranges. As suggested in the later text the variation between sensors were the major "noise".

I also wonder how this correlates to the later reconstruction due to the model variables as the authors commented in Line 117. If so, will normalizing the sensors' type help? Some related comments should be helpful to readers.

Eq (1) is formatted in a messy form, please fix it.

Line 144-145 - why the minimum leaf size was set to five? Due to five different sensor models? If so, please clarify it in the text.

Line 154-155 - what are the hidden layers and bias functions? Clarification is needed since this topic is exposed to a wide range of readership.

Line 158 - the Bayesian Regularization was shown to perform well for regression applications independent of network architecture - how this is correlated to this study specifically? Please make the argument of using any models more specific to the study here. I find a lot of statements too generic.

Figure 2 is very poorly represented. Fonts are too small and no clear indication of what Y-axis' are. Physical meaning of the parameters needs to be labeled clearly. The caption also requires more detailed descriptions instead of a generic description. This makes it hard for readers to get any meaningful interpretations themselves.

Figure 3 fonts need to be fixed. Too small and very unprofessional.

In Figure 3, only one day was picked as an example. How about other days that might have a higher variation/lower variation? Authors can consider placing some similar zoom-in comparisons into SI at least and comment on the observations. Especially with plots that have spikes of CH4, which are considered to have more meaningful data.

In Figure 4, the error bars are so large that the data in Z1 is essentially overlapping with the data in Z2. I understand that trend-wise, Z1 indeed showed a decreasing trend. Some necessary comments are needed to convince the readers.

More discussions should be laid out to illustrate the reason why the RF model is superior to the other two, on top of the supporting data.

Figure 6. Formatting needs to be fixed too.

A false response rate of 35 percent sounds pretty large, what is the metric on a golden standard? Authors should address that and compare their method to it.

Mobile monitoring is a big-picture merit of the authors' techniques. I also suggest a comparison of such methods to other stationary methods. Pros and cons?

Comments on the Quality of English Language

Okay.

Author Response

Reviewer #2:  This work aims to use low cost sensors for mobile monitoring of CH4 with different statistical models for calibration and peakfinding. I have listed a couple of comments for authors to address for revision –

We appreciate the reviewer’s comments, which contribute to improving this manuscript. We have considered each comment carefully and given responses below.

(1)Comment: A very fundamental question - can authors justify why they don't use the same model's of sensors for the same gas detection? For example, for methane, they have Figora and Sparkfun which have a very large mismatch in the sensing dynamic ranges. As suggested in the later text the variation between sensors were the major "noise".

Response: An explanation of why specific sensors were chosen for this project has been added to lines 105-109.

(2)Comment: I also wonder how this correlates to the later reconstruction due to the model variables as the authors commented in Line 117. If so, will normalizing the sensors' type help? Some related comments should be helpful to readers.

Response: An explanation of the model terms is given in lines 159-161. Additionally, though the variability of each sensor may be different, comparing the relative importance of each term in the RF model (Figure 5) to the range of each of the study variables (Table S2) shows that there is no correlation between the range of raw sensor values and the overall importance of each fitting parameter in our RF model.

(3)Comment: Eq (1) is formatted in a messy form, please fix it.

Response: The form of equation one has been fixed.

(4)Comment: Line 144-145 - why the minimum leaf size was set to five? Due to five different sensor models? If so, please clarify it in the text.

Response: The minimum leaf size was set to five as a leaf size of 5 minimized the RSMD on testing data. An explanation of this has been added to section 2.3 and relevant supporting statistics are included in supplementary Table S12.

(5)Comment: Line 154-155 - what are the hidden layers and bias functions? Clarification is needed since this topic is exposed to a wide range of readership.

Response: Additional information regarding the hidden layers and bias functions have been added to section 2.2 to clarify the methodology employed in this study. For additional clarification, an additional figure displaying the ANN architecture has been included in supplemental figures (Figure S6).

(6)Comment: Line 158 - the Bayesian Regularization was shown to perform well for regression applications independent of network architecture - how this is correlated to this study specifically? Please make the argument of using any models more specific to the study here. I find a lot of statements too generic.

Response: In this section we added several lines explaining why the Bayesian regularization works well for these applications, and tested several other regularization schemes which are included in the supplemental materials (Table S10).

(7)Comment: Figure 2 is very poorly represented. Fonts are too small and no clear indication of what Y-axis' are. Physical meaning of the parameters needs to be labeled clearly. The caption also requires more detailed descriptions instead of a generic description. This makes it hard for readers to get any meaningful interpretations themselves.

Response: We have edited Figure 2 accordingly.

(8)Comment: Figure 3 fonts need to be fixed. Too small and very unprofessional.

Response: We have altered the fontsize for Figure 3.

(9)Comment: In Figure 3, only one day was picked as an example. How about other days that might have a higher variation/lower variation? Authors can consider placing some similar zoom-in comparisons into SI at least and comment on the observations. Especially with plots that have spikes of CH4, which are considered to have more meaningful data.

Response: An additional figure has been added (S7) showing the same models with a different date displaying greater variability.

(10)Comment: In Figure 4, the error bars are so large that the data in Z1 is essentially overlapping with the data in Z2. I understand that trend-wise, Z1 indeed showed a decreasing trend. Some necessary comments are needed to convince the readers.

Response: While the errorbars are large for Figure 4 to the point that Z1 and Z2 may overlap, the t-test data (Table S8) which is referenced shows that there is a statistically significant difference between the mean RMSD values in Z1 and Z2. We added a citation to Table S8 in the text to further support our analysis.

(11)Comment: More discussions should be laid out to illustrate the reason why the RF model is superior to the other two, on top of the supporting data.

Response: We have added some additional details supporting why the RF model performance may be enhanced compared to other models (Lines 247-253).

(12)Comment: Figure 6. Formatting needs to be fixed too.

Response: We have fixed the formatting on Figure 6.

(13)Comment: A false response rate of 35 percent sounds pretty large, what is the metric on a golden standard? Authors should address that and compare their method to it.

Response: As this is a novel technique, there is not a ‘golden standard’ in the literature to suggest as a baseline comparison. However, we emphasize that this may be best suited for community monitoring or other low-cost applications as a screening tool rather than as a reference instrument. The initial interest from a local regulatory agency was to have a screening tool that could be broadly employed on all regulatory vehicles; the high value would be followed up with a standard reference instrument. 

(14)Comment: Mobile monitoring is a big-picture merit of the authors' techniques. I also suggest a comparison of such methods to other stationary methods. Pros and cons?

Response: A discussion of the pros and cons of this technique compared to stationary monitoring have been added to the conclusion section of the manuscript (Lines 332-342).

Reviewer 3 Report

Comments and Suggestions for Authors

The manuscript focuses on the development of mobile monitoring of fugitive methane emissions stemming from well sites in eastern Colorado using a range of low-cost sensors (XPOD) along with a reference-grade methane monitor. Materials and methods are presented in sufficient detail, original and calculated data are given in the supplementary materials, which makes it possible to assess the reliability of the data obtained. However, before accepting the article, some comments need to be addressed.

1. the function y is not fully displayed on line 136, it should be fixed.

2. The time of data collection is quite limited; can the authors assume whether the proposed model will work, for example, in winter?

3. In conclusion, it is advisable to provide explanations about the advantages and disadvantages of the proposed model of monitoring, as well as options for possible further research.

Author Response

The manuscript focuses on the development of mobile monitoring of fugitive methane emissions stemming from well sites in eastern Colorado using a range of low-cost sensors (XPOD) along with a reference-grade methane monitor. Materials and methods are presented in sufficient detail, original and calculated data are given in the supplementary materials, which makes it possible to assess the reliability of the data obtained. However, before accepting the article, some comments need to be addressed.

We appreciate the reviewer’s comments, which contribute to improving this manuscript. We have considered each comment carefully and given responses below.

(1)Comment: the function y is not fully displayed on line 136, it should be fixed.

Response:  Equation one has been corrected.

(2)Comment: The time of data collection is quite limited; can the authors assume whether the proposed model will work, for example, in winter?

Response: Ideally, data collection would have occurred over a full calendar year to ensure seasonal effects in methane concentrations as well as variation in environmental variables (including temperature and humidity) are fully accounted for. As temperature, humidity, and raw sensor values are all incorporated into the model, we may expect that for values of these variables close to the model, we may still trust our result with high fidelity. Along these lines, the model was able to perform well on unseen testing data.  However, using RF, ANN, and other machine learning models for extrapolation is often a poor choice, and any results from such an extrapolation should be used cautiously. We have addressed the short colocation period in the conclusion of this manuscript (Line 341-342).

(3)Comment: In conclusion, it is advisable to provide explanations about the advantages and disadvantages of the proposed model of monitoring, as well as options for possible further research.

Response: A discussion of the pros and cons of this technique compared to stationary monitoring as well as directions for future research have been added to the conclusion section of the manuscript (Lines 332-342).

Reviewer 4 Report

Comments and Suggestions for Authors

This manuscript details the utilization of a commercial metal oxide gas sensor array for methane identification making it of interest to a wide readership. The robust discussion and conclusions are well-founded and supported by the results. I recommend its publication in Sensors following the revision/clarification of the issues outlined below:

1. Clarification is needed for the term (XPOD). Please provide a detailed explanation.

2. Equation 1 is incomplete. Kindly correct and ensure its accuracy.

3. The methodology for feature extraction from the gas sensor array is unclear. Authors have utilized the varying resistances of gas sensors between the baseline and detected gases as inputs to the artificial neural network (ANN)? It seems to be all data with no  baseline gas (Clean air for reference). How to know that it’s methane since these sensors own cross sensitivity?

4. Please specify whether all data were used and include information on the sampling rate and the number of data points in the manuscript.

5. The limit of detection should be included.

6. The limitation of this model to predict the methane should be included. 

7. It’s well known that the TGS gas sensor strongly affects on the humidity. Based on the presented results, it’s not clear how to correct the humidity effect on this work.

8. All important components of device and real photograph of device should be included in the main manuscript.

Comments on the Quality of English Language

Minor editing of English language required

Author Response

This manuscript details the utilization of a commercial metal oxide gas sensor array for methane identification making it of interest to a wide readership. The robust discussion and conclusions are well-founded and supported by the results. I recommend its publication in Sensors following the revision/clarification of the issues outlined below:

We appreciate the reviewer’s comments, which contribute to improving this manuscript. We have considered each comment carefully and given responses below.

(1)Comment: Clarification is needed for the term (XPOD). Please provide a detailed explanation.

Response:Additional information has been included to describe the XPOD monitoring system (Lines 100-102).

(2)Comment: Equation 1 is incomplete. Kindly correct and ensure its accuracy.

Response: Equation one has been corrected accordingly.

(3)Comment: The methodology for feature extraction from the gas sensor array is unclear. Authors have utilized the varying resistances of gas sensors between the baseline and detected gases as inputs to the artificial neural network (ANN)? It seems to be all data with no  baseline gas (Clean air for reference). How to know that it’s methane since these sensors own cross sensitivity?

Response: For this analysis, we employed sensors with and without scrubbers to remove cross-sensitive species (TGS2611-E00,TGS-2611C00) and general VOC sensor (TGS 2602) to rule out measuring other gas phase hydrocarbons (Table 1). As our methane sensors with and without scrubbers were highly correlated, we can assume our signal was not due to cross-sensitive species. Additionally, as the general VOC sensor showed poor correlation with methane, and all other targeted sensors displayed higher correlations (Figure 2), we conclude that we are mostly measuring methane and not another gas-phase hydrocarbon.

(4)Comment: Please specify whether all data were used and include information on the sampling rate and the number of data points in the manuscript.

Response:The sampling rate and number of data points have been included in the manuscript (Lines 102-103, 144-147).

(5)Comment: The limit of detection should be included.

Response: This analysis does not focus on the limit of detection of the device, and additional analysis beyond the scope of the paper would be required to assess the limit using this methodology. The limit of detection is dependent upon the variables used, including temperature, humidity, and the reference methane data, and may not be generalizable so we did not focus our analysis on this metric. However, Figure 6 illustrates that the RF model is able to identify methane peaks on the single PPM scale.

(6)Comment: The limitation of this model to predict the methane should be included. 

Response: Several of the limitations of this technique were added to the conclusion section of the manuscript (Lines 332-342).

(7)Comment: It’s well known that the TGS gas sensor strongly affects on the humidity. Based on the presented results, it’s not clear how to correct the humidity effect on this work.

Response: Humidity included as an term in all models to correct for its effects on gas sensors. This is displayed in Equation 1.

(8)Comment: All important components of device and real photograph of device should be included in the main manuscript.

Response: To reduce the number of figures, photographs of the device and field setting were included in the supplementary materials. Additional information regarding the device was included in the materials and methods section to better characterize the XPOD monitor.

Round 2

Reviewer 1 Report

Comments and Suggestions for Authors

I have no extra comments.

Reviewer 2 Report

Comments and Suggestions for Authors

Concerns have been addressed accordingly.